# A Hierarchical Genotyping Framework Using DNA Melting Temperatures Applied to Adenovirus Species Typing

**DOI:** 10.3390/ijms23105441

**Published:** 2022-05-13

**Authors:** Ben Galvin, Jay Jones, Michaela Powell, Katherine Olin, Matthew Jones, Thomas Robbins

**Affiliations:** bioMérieux Inc., Salt Lake City, UT 84108, USA; jay.jones@biomerieux.com (J.J.); michaela.powell@biomerieux.com (M.P.); katherine.olin@biomerieux.com (K.O.); matthew.jones@biomerieux.com (M.J.); tom.robbins@biomerieux.com (T.R.)

**Keywords:** genotyping, DNA melting curve analysis, adenovirus, epidemiology, surveillance, Bayesian, BioFire, bioMérieux

## Abstract

Known genetic variation, in conjunction with post-PCR melting curve analysis, can be leveraged to provide increased taxonomic detail for pathogen identification in commercial molecular diagnostic tests. Increased taxonomic detail may be used by clinicians and public health decision-makers to observe circulation patterns, monitor for outbreaks, and inform testing practices. We propose a method for expanding the taxonomic resolution of PCR diagnostic systems by incorporating a priori knowledge of assay design and sequence information into a genotyping classification model. For multiplexed PCR systems, this framework is generalized to incorporate information from multiple assays to increase classification accuracy. An illustrative hierarchical classification model for human adenovirus (HAdV) species was developed and demonstrated ~95% cross-validated accuracy on a labeled dataset. The model was then applied to a near-real-time surveillance dataset in which deidentified adenovirus detected patient test data from 2018 through 2021 were classified into one of six adenovirus species. These results show a marked change in both the predicted prevalence for HAdV and the species makeup with the onset of the COVID-19 pandemic. HAdV-B decreased from a pre-pandemic predicted prevalence of up to 40% to less than 5% in 2021, while HAdV-A and HAdV-F species both increased in predicted prevalence.

## 1. Introduction

Characterizing genetic variation in pathogen populations and expanding the understanding of how they evolve between hosts can have important implications for public health [1]. Providing increased taxonomic information about a pathogen can inform medical decision-making. For example, it has been shown that a specific strain of Enterovirus (D-68) presents an increased risk of severe respiratory infections for some at-risk populations (e.g., pediatric and asthmatic patients) compared to other similar Enterovirus strains [2,3]. When this increased taxonomic resolution of a pathogen is available and provided in real-time, clinicians and health care professionals can make decisions for local and regional institutions to mitigate the effects of an outbreak [4].

At the present time, next-generation sequencing (NGS) is the primary method of genetic characterization and is increasingly being integrated into the clinical microbiology laboratory, but there still exist many challenges in its routine use, namely cost and time to results [5]. Alternatively, lab-developed tests (LDTs) to identify specific strains are common practice in diagnostics for the clinical microbiology laboratory. However, these tests cannot be distributed directly from lab to lab, and labs are not required to standardize assays used for the identification and reporting of specific strains. Commercially available diagnostic tests for common pathogens are widely available due to their low cost, time to result, and/or clinical accuracy; however, most do not have the ability to provide speciation of the identified pathogen.

One commercially available platform that is in wide use is the BioFire^®^ FilmArray^®^ system, a multiplex, real-time PCR diagnostic system from bioMérieux, Inc. (Salt Lake City, UT, USA). The BioFire system addresses many of the issues with NGS, LDTs, and many of the low-plex front-line testing options, in that the assays are standardized across labs using the BioFire system. The BioFire system is capable of providing a sample-to-answer result in about an hour, is in routine use in many labs, and is relatively low cost. Here, we propose a mathematical framework that takes advantage of the design of the molecular assays in the BioFire system and the known genetic variability within assay target regions to provide additional taxonomic resolution for a pathogen.

The BioFire system combines multiple assays in a single diagnostic test to report the presence or absence of multiple pathogens in a single patient sample. For each pathogen, there is at least one assay in the test that targets a specific region of its genome. However, there are some pathogens where multiple assays are necessary to capture strain diversity. The BioFire system uses PCR to amplify the target regions of a pathogen genome for each assay and an end-point melting curve analysis (MCA) of the amplicon for the detection and identification of the target [6]. As part of MCA, the BioFire system calculates the temperature of the maximum rate of disassociation of the double-stranded DNA. This metric is referred to as the melting temperature or Tm value [7,8]. A theoretical Tm value for each expected amplicon sequence can be computed with in silico analysis [9,10]. Here, we propose a mathematical framework to provide predicted speciation based on genotypic differences using the BioFire test result and the observed Tm values from well-characterized isolates, their theoretical Tm values, and the associated taxonomic lineage as training inputs.

As a practical application of this framework, we predict human mastadenovirus (HAdV) species from positive test results on the BioFire^®^ Respiratory 2.1 Panel and BioFire^®^ Respiratory 2 Panel, referred to herein as BioFire Respiratory Pathogen Panels or RPP. Adenoviruses are DNA viruses from the family *Adenoviridae*, typically associated with respiratory and gastrointestinal infections [11,12]. In some cases, such as in young children or the immunocompromised, these infections can be serious [13,14]. There are seven known species of HAdV (A–G) [15,16]. Of these species, HAdV (A–F) are most often associated with respiratory infections, with C resulting in more mild outcomes compared to B and E [17]. HAdV-G has only been observed to cause gastrointestinal infections. Outbreaks of HAdV are prone to occur in crowded community settings such as dormitories and hospitals [18]. While vaccines are effective at preventing outbreaks of infection caused by HAdV, they are currently only administered in select settings and are not widely available to the public [19]. HAdV typing in the United States is primarily motivated by the need to determine patterns of circulation, detect outbreaks, and inform vaccine and test design [20,21].

The RPP are designed to detect the six HAdV species known to cause respiratory infections in humans [22]. To ensure inclusivity, these products contain five assays that provide complementary coverage of this range of HAdV species. The RPP do not, however, distinguish between the HAdV species in the diagnostic result, instead returning a generalized Adenovirus “Detected” or “Not Detected” result.

In order to monitor circulation and detect outbreaks of HAdV, the model developed here is applied to the BioFire Syndromic Trends (Trends) system [23]. Trends is a network that collects and stores aggregated, deidentified patient test data from a subset of BioFire systems. More than 100 sites across the globe have contributed BioFire test data to Trends since 2015, with the aggregated dataset containing over 2 million RPP test results. By applying the HAdV speciation method to this dataset, we estimate changes in the circulation patterns of HAdV species from 2018 through 2021.

In the following sections, we detail the performance of an HAdV classification model. The model is then applied to RPP tests with Adenovirus “Detected” results in the Trends dataset to estimate HAdV species prevalence. The use of this method to augment HAdV typing efforts and the generalizability of this method to other genotyping applications are discussed. Finally, we define the formal mathematical framework used for predicting genotypic variability based on expected Tm values and detail how the model to classify HAdV species was optimized using a labeled dataset of well-characterized isolates.

## 2. Results

### 2.1. Hierarchical Classification Model Performance

A confusion matrix of the predicted HAdV species against known HAdV species using the optimized classification model is shown in Table 1. Values along the diagonal indicate that the prediction agreed with the known genotype for the sample, and off-diagonal elements indicate misclassifications. Table 2 summarizes the performance results of the model on the labeled dataset with the mean of the precision, recall, and F1-score across the ten cross-validation folds, along with support by HAdV species. The mean class-wise precision for HAdV-A of 0.88 indicates that when HAdV-A is predicted, it is correct 88% of the time. The mean class-wise recall for HAdV-A of 0.96 indicates that when the true classification for a sample is HAdV-A, the species is correctly predicted 96% of the time. The mean class-wise F1-score for HAdV-A of 0.92 indicates balanced accuracy as the harmonic mean of precision and recall. Support indicates the number of samples of each species included in the analysis (see the Methods section for more details of the training dataset). The macro average for each metric is the average across all classes with equal class weights and the weighted average is the average across all classes weighted by the number of samples in each class [24].

The logistic regression model finds coefficients for each assay’s posterior probabilities that optimize the model’s classification performance on the training dataset. In contrast, the assay design-informed model used for a sensitivity analysis sets the coefficients by the heuristic of expected assay reactivity and demonstrates 75% overall class accuracy for the same training dataset. The detailed performance results and confusion matrix for this method are shown in Appendix B, Table A1 and Table A2, respectively.

### 2.2. Application of Adenovirus Classification Model to Syndromic Trends

Figure 1A shows the optimized classification model applied to the RPP runs in Trends with an Adenovirus “Detected” result. The rate of Adenovirus “Detected” results are displayed as a rate of total RPP runs in the database. Subtypes are displayed in the stacked area under the total detections, indicating the percentage of RPP runs with an Adenovirus “Detected” result attributed to each species. Figure 1B shows the normalized percentage of each HAdV species from Figure 1A.

## 3. Discussion

### 3.1. Hierarchical Classification Model Performance

The mathematical framework for predicting genotypic differences applied to HAdV species typing showed robust cross-validation performance in assigning correct species labels to samples tested with the RPP, with an overall class accuracy of 95% and a standard deviation of 4% across the 10-fold cross-validation. However, the optimized classification model performance was not equal across all HAdV species with the lowest species class accuracy equal to 87% (see Table 2). Higher class precision and recall were observed for HAdV species B and C compared to A, D, E, and F. This may be attributed to the assay coverage of these species, as A and F and D and E often shared high posterior probabilities in similar temperature ranges. The model also demonstrates performance in line with the Xu, McDonough, and Erdman HAdV species typing assay that show an overall class accuracy of 96% with the lowest species class accuracy of 83% [25].

Compared to the optimized classification model, the assay design-informed classification model had two primary deficiencies in performance. The most discrepant results were observed with HAdV-A and HAdV-F samples, often mistaking one for the other. It also misclassified many HAdV-E samples as HAdV-D. While the optimized classification model exhibits the worst performance with HAdV-A and HAdV-F, it avoids most of the specific biases of the assay design-informed model by using the labeled data to inform the posterior probability weights. Although the training dataset used for these species classification models was comprised of a diverse set of serotypes for each HAdV species, we would expect lower accuracy than what is demonstrated here when applied to unseen serotypes.

### 3.2. Generalization of the Genotyping Framework

The mathematical framework presented here was well-suited for the RPP HAdV assays. These assays were designed specifically to capture the diversity of the HAdV genome across the different species. However, it could be generalized to classify genotypic diversity using other assays, provided there is variability in Tm values inside the relevant temperature range due to sequence differences of the amplicon sequences associated with genotypic diversity. The variation in Tm values associated with genotypic diversity must be greater than the temperature resolution of the instrumentation and the run-to-run variability for a reactive sequence. The error of the predicted Tm values for a given set of reaction conditions and amplicon sequences must also be less than the differences in Tm values between the genotypic groups. If assays were designed to maximize the variation of Tm values from different genotypic groups, the performance of this method could be further improved.

### 3.3. Adenovirus Surveillance

The relative prevalence of HAdV species in specific subpopulations has been studied extensively [11,12,18,19,26,27,28,29]. The surveillance of HAdV species in the greater population is currently addressed by the CDC’s National Adenovirus Type Reporting System (NATRS) [30]. This program collects and summarizes typing results from labs and hospitals and publishes a static prevalence estimation for several time periods. NATRS has published prevalence data from 2003 to 2016 collected from 11 sites and from 2014 to 2017 collected from at least 33 sites. We applied the HAdV species classification model to the RPP patient test data collected by the Trends system from 2018 through 2021. Because HAdV species prevalence varies from year to year, and the data collection timeframe does not overlap, no direct comparison of prevalence estimated here is made to that of NATRS [14,29]. However, the relative species proportions between NATRS (2014–2017) and the model predictions of the Trends data (2018–2020) are similar with HAdV-B and HAdV-C combined, consisting of greater than 80% of HAdV relative prevalence in both estimates. The application of the HAdV species classification model to Trends data supports and augments the surveillance efforts of NATRS. The Trends database contains a large pool of “Adenovirus Detected” results (>25,000) reported from over 100 testing sites across the United States. The application of the model to an active reporting system, such as Trends, confers several benefits over traditional passive reporting. First, this application requires minimal additional effort beyond running the samples on the RPP. Second, because all results are produced by the same testing platform, they share the same sample preparation protocol, reagents, and assays, which improves data consistency. Finally, because the data are automatically processed and categorized by the system and available in near-real-time, they can be used to detect species-specific outbreaks and monitor seasonality and changes in circulation patterns. The HAdV species classification model can be continuously applied to Trends data moving forward, and as more sites are added to the platform, more detailed patterns of circulation can be discerned.

Trends data were utilized in this study to investigate United States HAdV species prevalence. While Trends data has been used to suggest pathogen circulation dynamics [4,31,32], it should be noted that the testing practices and demographics of the institutions contributing to Trends are unknown. Biases in sites contributing to the network may exist that could give Trends a skewed representation of population pathogen prevalence. These limitations need to be kept in mind when making inferences about HAdV species prevalence based on the classification model results applied to Trends data.

Investigating the predicted prevalence data produced by the model shown in Figure 1A reveals several interesting results. Immediately apparent is the large drop in positivity in the spring of 2020 due to the COVID-19 pandemic. This pattern was not limited to HAdV, however, and was observed across all positivity in the Trends network [32]. As social practices return to normal over the course of 2021, rates of Adenovirus “Detected” results increase in Trends to levels seen prior to the onset of the pandemic. It is interesting to note in Figure 1B that the estimated proportion of HAdV-B does not recover to its previous state of about 30% of Adenovirus “Detected” results after the onset of the pandemic; in 2021 it has consisted of less than 5% of total Adenovirus “Detected” results. In the same timeframe, HAdV-C increased in relative prevalence to make up approximately 70% of all Trends Adenovirus “Detected” results. Additionally, two species (HAdV-A and HAdV-F) appear to be increasing in prevalence in the absence of HAdV-B. HAdV-A has increased from a low in 2020 of 4.5% relative prevalence to a high in 2021 of 11.6%, and HAdV-F increased from a low of 1.8% in 2020 to a high in 2021 of 17.9%. As stated above, this data may not reflect the actual circulating patterns of HAdV in the general population. Although this is an interesting observation on potential changes in the prevalence of several HAdV species, further study and monitoring are needed to understand the dynamics of circulating virus populations after a major disruption in social practices and competition between similar species.

## 4. Materials and Methods

In this section, we detail the general mathematical framework for predicting genotypic differences as applied to predict HAdV species from an unknown sample. To develop the framework, we first determine the HAdV species reactivity of each RPP HAdV assay by finding a complementary alignment of the forward and reverse primer sequences for HAdV entries in the Nucleotide collection nr/nt sequence database [33]. The Tm values are predicted in silico for each expected resulting amplicon sequence. These estimated Tm values are used to form a probability density function for the Tm value of each species. We then employ Bayes’ Theorem to estimate the posterior probabilities of each HAdV species given a Tm value. The results for all assays, with their associated HAdV species posterior probabilities, are the training input to a data-driven, hierarchical classification model to predict the species of an unknown sample. The classification model was optimized and the performance assessed and validated using 10-fold cross-validation on a labeled dataset. The validated model was applied to Adenovirus “Detected” results from RPP test data in the Trends system to predict HAdV species prevalence in the United States from 2018 through 2021.

### 4.1. Estimating Assay Reactivity

For each RPP HAdV assay, forward and reverse primer sets were used as query inputs for a BLASTn search against the nr/nt database [33]. The results from these queries were filtered to HAdV hits. We identified a predicted 2249 amplicons in HAdV genomes, where an amplicon is bounded by primers in the correct orientation and less than 1000 bp in length. The amplicons are linked to their respective taxonomic lineage, including the HAdV species. The set of HAdV species that have at least one predicted amplicon for an assay is considered to be reactive for an RPP HAdV assay. Table 3 contains the predicted reactivity of HAdV species for the five HAdV assays on the RPP.

The primer sequences for the RPP HAdV assays are not disclosed and the RPP HAdV assay names have been obfuscated for the commercial interest of bioMérieux, Inc.

### 4.2. Predicting Distributions of Tm Values

For a single HAdV assay on the RPP, let *S_i_* represent one of the HAdV species (A–F). For each species *S_i_*, there is a set of associated potential amplicon sequences denoted *R_i(j)_* for *j =* 1, 2, 3, … *n_i_*, with *n_i_* being the number of distinct amplicon sequences that are associated with species *S_i_* and amplified by the HAdV assay. Note that if the species is not expected to react with an assay, then *n_i_* = 0.

For each amplicon sequence *R_i(j_**_)_*, the predicted Tm value, based on the base-pair sequence of the amplicon, is denoted *µ_Tm_*_,_*_Ri(j)_*. For simplicity, we use the model proposed by Howley et al. for all Tm value predictions [9]. To account for system variability due to slight changes in chemistry, instrument conditions, and sample makeup, the Tm value for each *R_i(j)_* is represented by a normally distributed random variable *T_Ri(j)_*:(1)$$T_{R_{i\left(j\right)}}\sim N\left(\mu_{Tm,R_{i\left(j\right)}},\sigma_{sys}^{2}\right)$$

In Equation (1), the variance of *T_Ri(j)_*, *σ^2^_sys_*, represents the measurement error of the PCR system and is set to 0.5 °C standard deviations for all sequences in the application based on empirical evidence [6]. The distribution of expected Tm values for all expected reactive sequences of species *S_i_* can be represented as a weighted sum of all the individual sequence distributions *T_Ri(j)_*:(2)$$T_{S_{i}}\sim \overset{n_{i}}{\underset{j=1}{\sum }}w_{j}*T_{R_{i\left(j\right)}}$$
where 0 ≤ *w_j_* ≤ 1 for all *w_j_* and *∑_j_w_j_* = 1. For this application, the weights *w_j_* are naively set to 1/*n_i_* for all *j*. For a given assay and species *i*, *T_Si_* is the random variable that describes the possible Tm values based on the reactivity of that assay with the individual sequences included from the species. The associated probability density function for *T_Si_* is *f_TSi_*.

### 4.3. Posterior Probabilities of a Genotype Group

To calculate the posterior probability of each species given a *Tm* value, *x*, we employ Bayes’ Theorem for each *S_i_*:(3)$$P\left(x\in T_{S_{i}}|x\right)=\frac{P\left(x|x\in T_{S_{i}}\right)*P\left(S_{i}\right)}{P\left(x\right)}$$
where *P*(*x|x* ∈ *T_Si_*) is the likelihood of the Tm value, *x*, given it is from species *S_i_*. For a Tm value, *x,* the conditional probability that the Tm value is associated with species *S_i_* can be written: (4)$$P\left(x|x\in T_{S_{i}}\right)=\int_{x-\varepsilon }^{x+\varepsilon }f_{T_{S_{i}}}\left(x\right)dx$$
where *ε* is defined as the minimum resolution of the *Tm* detection algorithm, set to 0.01 °C for the BioFire system. Note that to account for the possibility of a reactive sequence not included in *R_i_*, we incorporate a uniform distribution *S*_unknown_~*U*(*Tm*_low_, *Tm*_high_), where *Tm*_low_ and *Tm*_high_ are the lower and upper bounds of the melting curve analysis range for all RPP HAdV assays, set to 75 °C and 95 °C, respectively. The final set of likelihood probabilities, *S*_all_, is the union of the conditional probabilities for an observed Tm and the uniform distribution *S*_unknown_.

The prior, *P*(*S_i_*), is a discrete distribution indicating the relative prevalence of HAdV species and for this application is set such that all species are equally likely. *P*(*x*) is the marginal probability calculated using the Law of Total Probability.

As an illustration of the methods, Figure 2 shows the progression from the estimated Tm values of expected reactive sequences (Figure 2A) to the likelihood distributions (Figure 2B) and subsequent posterior probability distributions (Figure 2C) for the RPP HAdV-5 assay. For a Tm value, each posterior probability describes the probability that the sample is of a particular species. Note that for each Tm value in the melting curve analysis range, the posterior probabilities for all species sum to one. Analogous figures for the other four HAdV assays (RPP HAdV-1, RPP HAdV-2, RPP HAdV-3, and RPP HAdV-4) can be found in Appendix A, Figure A1, Figure A2, Figure A3 and Figure A4.

### 4.4. Hierarchical Classification Model

The previous methods are applied to the five HAdV assays in the RPP independently to compute the posterior probabilities for the entire melting curve analysis range. These probabilities are combined into a data-driven hierarchical classification model to increase accuracy in predicting the species for an Adenovirus “Detected” result. Labeled data were generated by running isolates of known HAdV species using the RPP. A total of 347 RPP tests with an Adenovirus “Detected” result were sourced from three internal BioFire studies used to characterize RPP performance: the limit of detection study, the inclusivity study, and the cross-reactivity study. The results of these studies contribute to the instructions for use of the RPP [22]. The number of replicates, serotypes, and concentrations of the samples for each study are summarized in Table 4. This set of varied serotypes for each species, with multiple replicates of each isolate, provides a diverse training dataset for optimizing a data-driven classification model.

For each sample in the labeled dataset, the posterior probabilities for each assay were computed from the Tm value observed from each assay. Note that the Tm values for all data in this application have been normalized against the internal controls of the RPP [34]. The posterior probabilities are combined to create a feature matrix *X*, and the known species from each isolate form a response vector *y*. *X* and *y* are used to optimize a multiclass elastic-net regularized logistic regression model to predict HAdV species. This model was optimized using an L1 ratio of 0.6 with balanced class weights, and all other parameters set to the default values in the scikit-learn v1.0.1. package with Python v3.8 [35]. The performance of the optimized classification model was assessed and validated using 10-fold cross-validation and reported as the mean and standard deviation of the accuracy across all folds. The mean of the precision, recall, and F1-scoring across the ten cross-validation folds were also used to assess overall and class-wise performance [24].

In addition to the optimized logistic regression model, we developed another HAdV species classification model as a sensitivity analysis to ensure that the data-driven model provided increased performance over a naive approach. This model is referred to as the assay design-informed classification model and sets predetermined weights for each of the posterior probabilities based on a priori knowledge of assay reactivity. The weights for the posterior probabilities are set equally across all species that an assay has expected reactivity to, as shown in Table 1. For each sample, the prediction is determined by computing the Argmax of the weighted sum of posterior probabilities [35]. Because this model was not optimized using the labeled dataset, no cross-validation was performed.

### 4.5. Application to Syndromic Trends

The optimized classification model was elected to be used for application to Trends due to its increased performance over the assay design-informed classification model developed for the sensitivity analysis. HAdV species prevalence was predicted from Adenovirus “Detected” results from RPP test data in the Trends system in the United States from 2018 through 2021.

## 5. Conclusions

Using a priori information and in silico analysis, we developed a mathematical framework for providing additional taxonomic resolution for pathogen targets of the highly multiplexed BioFire system. This framework is flexible enough to provide increased genotypic classification for any PCR-based test with end-point melting analysis, so long as the genotypic variability being classified results in predicted Tm value variability. We applied this framework to the problem of predicting HAdV species from RPP Adenovirus “Detected” results. The HAdV species classification accuracy is enhanced by combining the results from multiple assays in a data-driven regularized logistic regression model. The model exhibited high accuracy on the labeled dataset. As a result, the optimized classification model was applied to unlabeled data from the Trends network. These results show a marked change in both the predicted prevalence for HAdV and the species makeup with the onset of the COVID-19 pandemic. In particular, HAdV-B decreased from a pre-pandemic predicted prevalence of approximately 30% to less than 5% in 2021, which led to relative increases in predicted prevalence for HAdV-A and HAdV-F during that same timeframe.

There are two areas where we would like to continue these efforts: the further characterization and validation of the species classification model and integrating the model into the Trends system for the real-time analysis of RPP HAdV-positive test results. During development, we trained the model on 347 samples covering 26 unique HAdV serotypes. Extending the dataset coverage to include more samples with varied serotypes from independent sources would provide additional confidence to the model’s predictions. Further development to integrate this model into the Trends system could augment CDC efforts to surveil the circulation of HAdV species and facilitate outbreak detection in real-time.

## Figures and Tables

**Figure 1 ijms-23-05441-f001:**
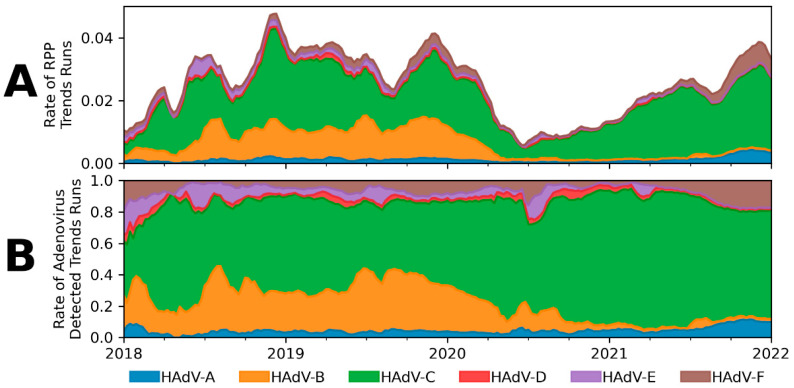
Estimated HAdV species prevalence from Trends data. HAdV species predicted from the optimized classification model applied to RPP Adenovirus “Detected” results in the Trends database from 2018 through 2021. Rates are calculated on a weekly basis and smoothed with a six-week centered rolling window. (**A**) Estimated prevalence of each HAdV species shown as a proportion of total runs of RPP. (**B**) Estimated prevalence of each HAdV species shown as a proportion of total Adenovirus “Detected” results.

**Figure 2 ijms-23-05441-f002:**
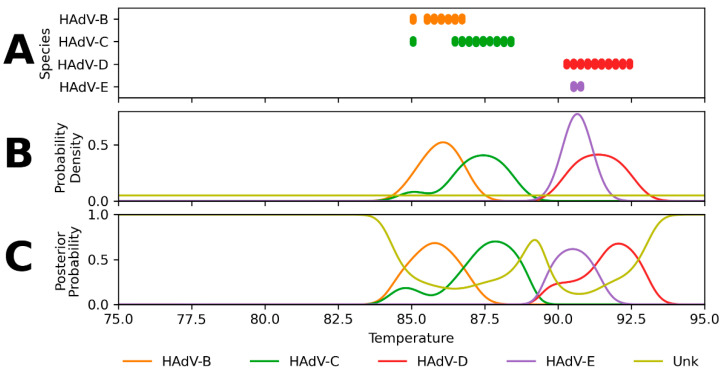
Predicted Tm values of reactive sequences, likelihood distributions, and resulting posterior probability distributions for reactive species with the RPP HAdV-5 assay. This assay had predicted reactivity with HAdV serotypes from the HAdV-B, HAdV-C, HAdV-D, and HAdV-E species. The assay shows the most predicted reactivity in the range of 85–93 °C. (**A**) Predicted Tm values for each expected reactive species. (**B**) Estimated likelihood distributions of Tm values. (**C**) Resulting posterior probabilities across the temperature range. Species HAdV-B, HAdV-C, HAdV-D, and HAdV-E have peak posterior probabilities at 85.8, 87.9, 92.1, and 90.5 °C, respectively.

**Table 1 ijms-23-05441-t001:** Confusion matrix of the predicted HAdV species by the optimized classification model against the true HAdV species.

HAdV Species		Predicted Label					
		A	B	C	D	E	F
Known Label	A	46	0	0	0	0	2
	B	0	81	0	0	2	0
	C	0	0	57	0	0	0
	D	0	0	0	53	4	3
	E	0	0	0	0	43	1
	F	6	1	0	0	0	48

**Table 2 ijms-23-05441-t002:** Detailed performance characteristics for the optimized classification model.

HAdV Species	Precision	Recall	F1-Score	Support
A	0.88	0.96	0.92	48
B	0.99	0.98	0.98	83
C	1.00	1.00	1.00	57
D	1.00	0.88	0.94	60
E	0.88	0.98	0.92	44
F	0.89	0.87	0.88	55
Macro average	0.94	0.94	0.94	
Weighted average	0.95	0.95	0.95	
Accuracy (±s.d.)	0.95 (±0.04)			

**Table 3 ijms-23-05441-t003:** The predicted HAdV species reactivity of the five RPP HAdV assays.

RPP HAdV Assay	HAdV Reactive Species
RPP HAdV-1	B, D, E
RPP HAdV-2	A, F
RPP HAdV-3	C
RPP HAdV-4	A, E, F
RPP HAdV-5	B, C, D, E

**Table 4 ijms-23-05441-t004:** A description of the labeled data included in the logistic regression classification model optimization. These studies contain different isolates of each HAdV species tested at varying concentrations.

HAdV Species	Limit of Detection Study		Inclusivity Study		Cross-Reactivity Study		Species Run Count
	Isolate Serotype	Run Count	Isolate Serotype	Run Count	Isolate Serotype	Run Count	
A	A18	39	A12	3	A31	3	48
			A31	3			
B	B7a	40	B3	3	B3	3	83
			B7d/d2	3			
			B7h	3	7a	3	
			B11	4			
			B14	3	B14	3	
			B16	3			
			B21	3	B21	3	
			B34	3			
			B35	3			
			B50	3			
C	C2	39	C1	3	C2	3	57
			C5	3	C5	3	
			C6	3	C6	3	
D	D37	40	D8	4	D8	3	60
					D20	3	
			D20	3	D26	3	
					D37	4	
E	E4a	38	E4	3	E4	3	44
F	F41	40	F40	6	F40	3	55
			F41	3	F41	3	
					Total Run Count		347

## Data Availability

All relevant data and code needed to reproduce the results of this work are included in the repository provided in the submission: https://bitbucket.org/ben_w_galvin/adenovirus-species-typing/src/master/.

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
