# Peer review of "A Hierarchical Genotyping Framework Using DNA Melting Temperatures Applied to Adenovirus Species Typing"

_ijms, 2022, doi:10.3390/ijms23105441_

Round 1

Reviewer 1 Report

Dear authors,

Your manuscript “A hierarchical genotyping framework using DNA melting temperatures applied to adenovirus species typing” is very interesting and gives an important and potentially useful tool for epidemiological and clinical studies of adenovirus. The manuscript is well-written, and I have only few remarks:

Lane 47 – “bioMerieux” – please, add information about country.

Lane 72 – “Adenoviridae” should be in italic.

Author Response

Thank you for your comments and time spent reviewing this manuscript. We have addressed your specific concerns in the attached document (indicated by comments). In addition to your specifically requested changes we adjusted the bibliography, acknowledgments, and conflict of interest statements as requested by comments in the document.

Reviewer 2 Report

Dear Authors,

This is a compelling manuscript describing an in silico design for AdV genotyping using a proprietary commercial PCR diagnostic platform for virus detection. The introduction and methods are well written and clear, even to a non-bioinformatician like myself. The conclusions are sound and highly informative, while the entire manuscript is polished. There is little editing required.

Action: I was not able to locate reference 34 - perhaps this reference isn't complete in the document.

Action: I would like to see this predictive in silico modelling tested experimentally going forward using a sample bank containing various confirmed AdV genotypes within each serotype. A small comment to this affect should be included in the conclusions, and would add weight to the validity of the in silico classification model.

This would be informative in light of the recent increased incidence of hepatitis in children in the UK and USA. The possibility of an association with a specific AdV genotype and/or SARS-CoV-2 is a compelling reason for the need to provide predictive serotyping information, especially during outbreaks.

Author Response

Thank you for your comments and time spent reviewing this manuscript. We have addressed your specific concerns in the attached document (indicated by comments). The missing reference was indeed an incomplete entry for a patent application. In addition to your specifically requested changes we adjusted the bibliography, acknowledgments, and conflict of interest statements as requested by comments in the document.

Your comment on confirmatory testing is in line with our own thoughts. While our own internal datasets contained 26 typed isolates and a total of 347 samples, testing with an independent dataset would lend credence to our argument. To this end, we are currently working with external collaborators to obtain additional labeled data (HAdV positive samples run with a BioFire RPP that have since been tested for serotypes) from their sample banks.

Finally, we are aware of the recent hepatitis outbreaks in the UK and US and are sending the CDC data from the Trends system to help with their investigations. We hope to continue to work with them in any way possible to aid in their response to these types of outbreaks.
